# Effect of Single Session of Swedish Massage on Circulating Levels of Interleukin-6 and Insulin-like Growth Factor 1

**DOI:** 10.3390/ijms25179135

**Published:** 2024-08-23

**Authors:** Ville Stenbäck, Inka Lehtonen, Kari Antero Mäkelä, Ghulam Shere Raza, Venla Ylinen, Rasmus Valtonen, Tuomas Hamari, Jaroslaw Walkowiak, Mikko Tulppo, Karl-Heinz Herzig

**Affiliations:** 1Research Unit of Biomedicine and Internal Medicine, Faculty of Medicine, University of Oulu, 90220 Oulu, Finlandghulam.raza@oulu.fi (G.S.R.);; 2Kontinkangas Unit, Educational Consortium OSAO, 90220 Oulu, Finland; 3Pediatric Gastroenterology and Metabolic Diseases, Pediatric Institute, Poznan University of Medical Sciences, 60572 Poznan, Poland; 4Biocenter Oulu, Medical Research Center (MRC), Oulu University Hospital, 90220 Oulu, Finland

**Keywords:** massage therapy, myokines, IL-6, IGF-1

## Abstract

Massage therapy increases muscle blood flow and heat, relieving pain, improving immune function, and increasing vagal activity. The mechanisms are unclear. Muscles release cytokines and other peptides called myokines. These myokines exert their effects on different tissues and organs in para-, auto-, and endocrine fashion. The aim of this intervention study was to investigate if massage therapy affects circulating myokine levels. A total of 46 healthy, normal-weight subjects (15 men) aged 18–35 were recruited. Forty-five minutes of massage Swedish therapy was applied to the back and hamstrings. Blood samples via cannula were taken at the baseline, during the massage (30 min), end of the massage (45 min), and 30 min and 1 h after the massage. Interleukin 6 (IL-6) and insulin-like growth factor 1 (IGF-1) were measured as surrogate markers by ELISAs. There was a significant increase in IL-6 from 1.09 pg/mL to 1.85 pg/mL over time (Wilks’ Lambda Value 0.545, *p* < 0.000; repeated measures ANOVA). Pair-wise comparisons showed a significant increase after 1 h of massage. No significant increase was observed in IGF-1 levels. The change in myokine levels was not correlated with muscle mass (*p* = 0.16, 0.74). The increase in IL-6 suggests that there might be anti-inflammatory effects, affecting glucose and lipid metabolism pathways via IL-6 signaling to muscles, fat tissue, and the liver.

## 1. Introduction

Massage therapy is a widely used tool to enhance recovery and alleviate muscle-originated pain caused by spasms and muscle tensions [1]. One of the most commonly used massage styles is called “Swedish massage”, which incorporates techniques such as effleurage (stroking), petrissage (kneading), tapotement (percussion), shaking/vibration, and cross-fiber friction (transverse friction). The massage relates back to the Swedish physiologist and gymnastics instructor Pehr Henrik Ling (1776–1839), which subsequently has been promoted worldwide as Western or Swedish massage [1]. It has been used for the recovery of skeletal muscle from strenuous exercise [2,3], different chronic diseases [4], pain relief, e.g., during labor [5], and balance performance [6].

The mechanisms of massage therapy are mainly based on increased blood flow to the massaged area, with increases in muscle and skin temperature [7,8]. The effects can be divided into mechanical and indirect effects. Mechanical effects include stretching of soft tissue, breaking scar tissue and fibrous adhesions, and increasing tissue elasticity and tissue permeability. Indirect effects are increased microcirculation [9,10]. Only a few studies have studied the physiological effects of massage therapy in a non-exercise setting in normal-weight, healthy subjects. A single session of Swedish massage has been shown to decrease cortisol and cytokine levels [11]. In more recent study by Do-Jin and colleagues, repeated weekly massage sessions for 5 weeks had the same effects as a single session massage, and those subjects who participated in two sessions weekly instead of one showed increased levels of oxytocin (OT), arginine-vasopressin (AVP), and adrenocorticotropin (ACTH) [12]. Massage therapy after exercise-induced muscle damage attenuated inflammatory cytokine levels, which may reduce pain via the same mechanisms as nonsteroidal anti-inflammatory drugs (NSAIDs) [13]. Diego and colleagues found a decrease in heart rate and an increase in overall relaxation measured by an electroencephalogram (EEG) [11]. In hypertensive subjects, blood pressure decreased after receiving a 30 min massage therapy twice per week over five weeks [14].

Skeletal muscles synthesize and release cytokines and peptides named myokines during and after physical activity [15,16]. These myokines exert their effects on different tissues and organs, including the liver, adipose tissue, bone, pancreas, and cardiovascular system in a para-, auto-, and endocrine fashion [17,18]. Myokines consist of interleukins (e.g., IL-6 and IL-15) and growth factors, such as IGF-1 and, fibroblast growth factor 21 (FGF-21), vascular endothelial growth factor A (VEGF-A), brain derived neurotrophic factor (BDNF), myostatin, and myonectin [19,20]. Exercise and other mechanical stimuli increase IL-6 and IGF-1 levels, produced by differentiated myotubes, osteocytes, and osteoblasts [21]. Resistance training, dependent on intensity and volume, induces hormonal responses that influence muscle development [22]. IL-6 affects lipid and glucose metabolism and insulin sensitivity and decreases inflammation [23,24]. IGF-1 secretion is stimulated by growth hormone in the liver [22], promoting growth and development [25]. Low IGF-1 levels are associated with several diseases, like type-2 diabetes mellitus (T2DM), cardiovascular diseases (CVDs), sarcopenia, osteoporosis, and frailty [26]. In the muscle, IL-6 and IGF-1 activate the anabolic Janus kinase/signal transducer and activator of transcription (JAK/STAT) pathway, which regulates many cellular processes, including proliferation, differentiation, adhesion, migration, invasion, organ development, and immunity functions [27]. In addition, the phosphatidylinositol-3-kinase and the mammalian target of rapamycin (PI3-K/AktT/mTOR) pathway is activated, affecting cellular processes in proliferation, growth, protein synthesis, and glucose and lipid metabolism and homeostasis with increased glucose uptake and fat oxidation [28,29]. Myokines are suggested to play a part in the signaling of health benefits of exercise and, therefore, act as “exercise factors”. The objective of the present study was to investigate the effect of a single session of Swedish massage on two myokines selected as surrogate markers. Our hypothesis was that the increased circulation, temperature, and mechanical activation of the muscle would cause an increase in these myokine levels.

## 2. Results

The BMIs of participants were on average 23.6 ± 2.3 (19.5–29.4); they weighed 68.5 ± 12.3 kg (47.5–100.6 kg) and were 169.8 ± 9.5 cm (152.0–190.0 cm) tall with a waist circumference of 77.5 ± 8.0 cm (64.5–94.0 cm) (Table 1). The mean skeletal muscle mass was 29.6 ± 7.2 kg (16.7–51.4 kg). The mean values for systolic and diastolic blood pressure were 121.3 ± 13.3 mmHg (86.0–155.5 mmHg) and 78.7 ± 8.6 mmHg (57.5–98.5 mmHg). All subjects were physically active and did not consume excessive amounts of alcohol.

IL-6 and IGF-1 concentrations before, right after (at 45 min), and after 60 min of massage therapy are shown in Table 2. Before the massage, mean IL-6 and IGF-1 concentrations were 1.10 ± 0.94 ng/mL and 157.3 ± 66.8 ng/mL, respectively (Figure 1). At the end of the 45 min massage, mean values were 1.20 ± 0.91 ng/mL for IL-6 and 159.8 ± 57.2 ng/mL for IGF-1. Sixty minutes after the end of the massage therapy, IL-6 values rose to 1.85 ± 1.30 ng/mL, while IGF-1 values were 159.0 ± 60.9 ng/mL (Figure 2). IL-6 concentrations 60 min after the massage were significantly different from the two earlier ones (F = 0.310, Wilks’ Lambda Value 0.545, *p* < 0.000). No differences were found in the IGF-1 levels (F = 18.36, Wilks’ Lambda Value 0.995, *p* < 0.893). IL-6 concentration increased significantly 60 min after the massage compared to the baseline and directly after massage (III) (95% confidence interval (CI) [−0.97, −0.38], *p* < 0.001 and 95% CI [−1.01, −0.51], *p* < 0.001, respectively). No effect was observed between other time points or IGF-1 (Table 2). The change in IL-6 level was not correlated with absolute muscle mass (0.211, *p* = 0.16).

## 3. Discussion

A 45 min bout of Swedish massage significantly increased IL-6 but not IGF-1 concentrations. The biggest increase was observed 1 hour after the massage therapy session (45 + 60 min from baseline). Surprisingly, the change was not dependent on the muscle mass of the participant. In another study consisting of a single session (45 min) of Swedish massage and light touch, the authors did not report any significant changes in IL-6 concentrations or differences in the changes between groups [30]. They found a decrease in AVP and cortisol (CORT) in the massage group and an increased number of lymphocytes, CD25 lymphocytes, CD56 lymphocytes, CD4  lymphocytes, and CD8 lymphocytes in blood (effect sizes from 0.14 to 0.43), indicating that the massage therapy may enhance the immunity by raising the number of circulating lymphocytes [30,31,32].

IGF-1 has been less studied in the context of massage therapy. Strength exercise and massage have been shown to increase IGF-1 levels [14]. Preterm infants gained more weight and had increased serum insulin and IGF-1 levels after three separate 15 min periods per day for 5 days of massage [33,34]. The higher weight gain in preterm infants has been contribute to the increased IGF-1 and decreased cortisol levels. It has been shown that chronic high cortisol levels inhibit growth hormone secretions [35].

IL-6 increases basal glucose uptake, lipolysis, and fat oxidation via activation of adenosine 5′-monophosphate-activated protein kinase (AMPK) [36]. The PI3-K/AKT signaling pathway activated by IL-6 regulates cellular functions that relate to the development of obesity and diabetes mellitus by affecting the translocation of GLUT 4 transporter and lipid metabolism via Forkhead box protein O1 (FoxO1), mammalian target of rapamycin complex 1 (mTORC1), and sterol regulatory element-binding proteins (SREBPs) [24,28]. Subjects with two massages per week had increased oxytocin levels and decreased vasopressin and cortisol levels, but little effects on lymphocyte markers and a slight increase in interferon-γ, tumor necrosis factor-α, interleukin IL-1β, and IL-2 levels, suggesting an increased production of pro-inflammatory cytokines compared to the other group. Differences in IL-6 levels were not significant [31]. The study suggests that repetitive massage therapy has cumulative effects on most measured markers.

Massage has several physiological effects and enhances immunity function. The effect was observed especially with moderate pressure massage [32]. Response to massage included hormonal changes in endorphins, serotonin, cortisol, oxytocin, cytokines, and myokines [31,37]. Massage therapy was associated with decreased cortisol, nitric oxide, and β-endorphin levels and increased oxytocin levels in subjects (n = 65) receiving a 15 min massage and a control group that rested for 15 min [38]. Oxytocin permits stress and anxiety-relieving effects, while cortisol mediates stress responses [39,40].

Our investigation has several strengths, including that all study subjects were healthy, living in the same area with a similar lifestyle. We had five blood drawing time points, which provided us with a wider possibility to follow the dynamics of the secreted markers. Body composition analysis was measured with bioimpedance enabling comparison between myokines and muscle mass. The limitations of our study were the inclusion of more females than men and that different massage therapy students performed massages on different subjects, which might slightly differ in style and pressure. To minimize this issue, all the massage therapy students were instructed to perform the massage therapy in the same way, and they performed the massage under supervision.

## 4. Materials and Methods

Participants: 46 healthy, normal-weight subjects (15 men) aged 18–35 from the University of Oulu and Educational Consortium OSAO were enrolled from e-mail registrations on a voluntary basis. Participants did not receive any compensation except information on their health gathered during the study. Exclusion criteria for the study were the following: pregnancy, lactation, underage or over 35 years, BMI less than 20 or over 28, diseases like chronic inflammatory, autoimmune, severe respiratory diseases, cardiovascular diseases, hypertension, any skeletomuscular diseases or disabilities, diabetes, celiac disease, or usage of drugs affecting the immune response. Subjects were asked not to do strenuous exercise and not to drink alcohol on the day before the study visit. The subjects were fasted and refrained from consuming caffeine and nicotine at least 10 h before the study visit, while drinking water was allowed. Height in centimeters and weight in kilograms were measured with one decimal accuracy. The BMI was calculated as weight (kg) divided by the square of height in meters (m^2^). Body composition was determined using bioimpedance with InBody 720 (Biospace, Co., Ltd., Seoul, Republic of Korea). Blood pressure and waist circumference were measured, and questionnaires were filled in by the participants before the massage. The questionnaires included medication, hormonal contraception, physical activity, smoking/snus, alcohol consumption (AUDIT; Alcohol Use Disorders Identification Test), and nutritional supplements. A cannula was inserted into the cubital or the cephalic vein to enable blood collection before, during, and after massage by a physician.

Swedish massage: Subjects participated in the morning between 8.00 and 11.00 in a single Swedish massage session lasting 45 min. The subjects carrying shorts laid supine on the massage table in a single booth in a warm room (22 °C) with background music. Typical treatments included effleurage (stroking), petrissage (kneading), tapotement (percussion), friction, and vibration. The massage was focused on the back, hamstrings, and calves to maximize muscle mass engagement and blood flow. Twenty-four massage therapy students (OSAO) performed the massage therapy under the close supervision of their teacher (registered under the National Supervisory Authority for Welfare and Health National Supervisory Authority for Welfare and Health operating under the Ministry of Social Affairs and Health, Finland), using consistent basic techniques and pressure. After 45 min, the subject stayed in the massage booth for an additional 60 min. No adverse events were recorded.

Measurements: Blood samples were collected (1) before the massage, (2) 30 min into the massage, (3) at the end of the massage (45 min), and after (4) 30 min and (5) 60 min of the massage. Blood samples were collected in BD Vacutainer K2E (EDTA) tubes and centrifuged (1700× *g*, 10 min at 4 °C) on the site after cooling the blood on ice immediately after collection. Plasma was divided into 1.5 mL Eppendorf tubes and frozen. IL-6 and IGF-1 were analyzed from the blood samples using commercial ELISA kits (HS600C, DG100B, Biotechne, R&D Systems Inc., Minneapolis, MN, USA) [41,42]. Intra- and interassay% CVs for IL-6 were 5.86 and 8.31, respectively, and for IGF-1 were 3.05 and 7.95, respectively. No cross-reactivity has been reported for either of the kits in physiological conditions. After preliminary analysis of pilot samples, only three time points were further analyzed because of the most significant differences: (1) Basal, (3) at the of the massage (45 min), and (5) after 60 min of massage.

Statistical analysis: All statistical analyses and graphs were drawn using IBM SPSS Statistics 27. Repeated measures ANOVA was used to analyze the myokine variables over time points. A paired samples *t*-test was used in the pairwise comparisons with Bonferroni correction. Linear Pearson correlation was determined to study the effect of muscle mass (absolute) to the changes in myokine levels. All the values are represented as mean ± standard deviation (mean ± SD).

## 5. Conclusions

A 45 min Swedish massage significantly increased circulating IL-6 levels after 1 hour of a single session. No significant changes were observed in IGF-1 levels. The change was not dependent on the muscle mass of the subjects. The rise in IL-6 levels could mediate anti-inflammatory effects and could potentially enhance glucose and lipid metabolism pathways via signaling to muscles, fat tissue, and the liver. The benefits of massage might be mediated via the release of myokines, which possibly depend on pressure, duration, and repetition of the massage.

## Figures and Tables

**Figure 1 ijms-25-09135-f001:**
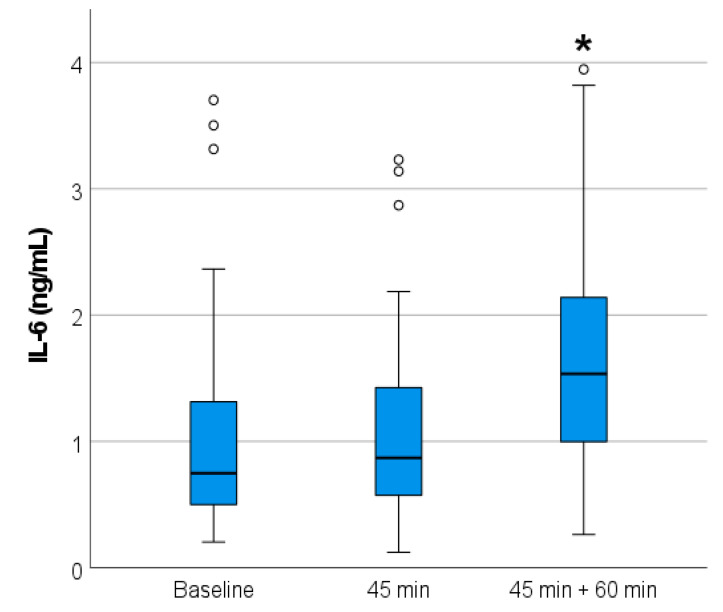
Interleukin 6 (IL-6) concentrations (ng/mL) in the subject’s plasma before, right after (45 min), and after 60 min of the massage therapy. * Significant difference between other time points (*p* < 0.01).

**Figure 2 ijms-25-09135-f002:**
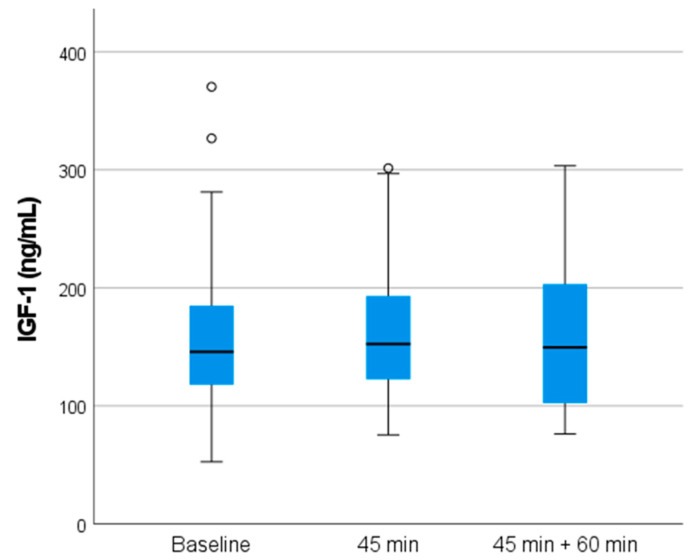
Insulin-like growth factor 1 (IGF-1) concentrations (ng/mL) in the subject’s plasma before, right after (at 45 min), and after 60 min of the massage therapy. No significant difference was observed between the time points (*p* < 0.893).

**Table 1 ijms-25-09135-t001:** Mean values and range of skeletal muscle mass (SMM), body mass index (BMI), weight, height, waist circumference, and systolic and diastolic blood pressure.

	Mean ± Std.	15 Men, 31 Women
SMM (kg)	29.6 ± 7.2	16.7–51.4
BMI	23.6 ± 2.3	19.5–29.4
Weight (kg)	68.5 ± 12.3	47.5–100.6
Height (cm)	169.8 ± 9.5	152.0–190.0
Waist (cm)	77.5 ± 8.0	64.5–94.0
RRSys	121.3 ± 13.3	86.0–155.5
RRDias	78.7 ± 8.6	57.5–98.5

**Table 2 ijms-25-09135-t002:** Interleukin 6 (IL-6) and insulin-like growth hormone 1 (IGF-1) mean values in three time points (before, right after (at 45 min), and after 60 min of massage therapy). Paired samples *t*-test values in comparison to other time points. I—baseline; II—45 min; III—60 min after the massage.

				Paired Samples *t*-Test		
IL-6		Mean ± Std.		Upper	Lower	Sig.
	I	1.10 ± 0.94	1–3	0.08	−0.24	0.312
	II	1.20 ± 0.91	1–5	−0.51	−1.01	<0.001
	III	1.85 ± 1.30	3–5	−0.38	−0.97	<0.001
**IGF-1**		Mean ± std.		Upper	Lower	Sig.
	I	157.3 ± 66.8	1–3	8.01	−12.95	0.637
	II	159.8 ± 57.2	1–5	10.57	−13.96	0.782
	III	159.0 ± 60.9	3–5	9.83	−8.28	0.863

## Data Availability

The original contributions presented in this study are included in the article. Further inquiries can be directed to the corresponding author/s.

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
