# Peer review of "Effect of Single Session of Swedish Massage on Circulating Levels of Interleukin-6 and Insulin-like Growth Factor 1"

_ijms, 2024, doi:10.3390/ijms25179135_

Round 1
Reviewer 1 Report
Comments and Suggestions for Authors
Thank you for inviting me as a reviewer of this valuable manuscript. This study presented results before and after one massage session. There is no control group and no follow-up check (e.g., at 1 week), which raises concerns about the objectivity of the research findings. I recommend following suggestions for improving the quality of manuscript.
Study Title
(Comment 1) Please specify the study design in the title.
- e.g. single group clinical stud
Introduction Section
(Comment 2) Authors mentioned that “Massage therapy is a widely used tool…” (line 31-32). I recommend authors to supplement information on globally recognized typical massages.
- Also, statistical information is recommended
(Comment 3) I recommend authors to add historical, statistical (e.g. utilization) and institutional information (e.g. insurance).
Method Section
(Comment 4) I recommend authors to reorganize the sub-titles and contents according to PICO (Participants, Intervention, Comparison, Outcome Measure). You can exclude the Comparison section from the sub-title.
- The intervention (treatment process) is unclear. Please provide detailed information on the Swedish massage procedure, supported by references. This is crucial for study reproducibility.
- Has the supervisor (teacher) received national certification? Additionally, have the twenty-four massage therapy students received training for clinical research procedures?
- Please provide a massage manual or protocol for clinical research as a supplementary file.
Result Section
(Comment 5) I recommend authors to include information about adverse events in the outcome measures.
(Comment 6) Regarding Table 2, please supplement explanations for I, III, and V in both the main text and at the bottom of the table.
Reviewer 2 Report
Comments and Suggestions for Authors
The article by Stenback et al. describes the effects of a massage scheme on two blood parameters, being IL-6 and insulin-like growth factor 1. The manuscript needs thorough revision in order to become acceptable for publication.
Firstly, the study is limited and therefore would, in my opinion, best be presented as a brief report. Secondly, the study evaluated the levels at 5 timepoints, but only presents 3 of those. Levels at all timepoints and for the individual subjects should be show in two graphs. Thirdly, significantly more women were included, which is odd for a study that recruits healthy young people and must have been spoiled for choice. Please comment.
References are not very recent, with only 3 of 28 dating to 2020 or more recent. Perform an extra literature search for relevant studies published in the last years, and discuss.
Corrections need to be made to the text. Ex. What does the sentence on line 50 “Skeletal muscles … [10].” signify? Line 53 Myokines consists; use of ‘like’.
Reviewer 3 Report
Comments and Suggestions for Authors
- Although normal IL-6 values do not differ between genders, the authors should differentiate the results in Table 1 by gender for a better understanding of the reader.
- There is an important bias in the authors' methodology by using 24 different massage therapists in their operational procedure. This point should be justified in more detail.
- Do the examples or Swedish massage given in the "Introduction" section exhaust the literature review in this area?
-Prior to providing the Results, it would be worth describing the Materials an Methods.
-I suggest analysing the literature in terms of publication date, includindg incresing the number of recent publications and limitins the number of publications from distant years
Round 2
Reviewer 1 Report
Comments and Suggestions for Authors
Authors have adressed most of my comments. I request the following two revisions.
- Regarding Comment 1, was this study approved by the IRB as a clinical study? If it is not a clinical study, please specify the study design.
- Authors did not adress Comment 3. Providing brief information about the intervention is very important.
(Comment 3) I recommend authors to add historical, statistical (e.g. utilization) and institutional information (e.g. insurance) of Swedish massage
Author Response
We thank the referee for time to review our manuscript. Please find our point to point response.
- Regarding Comment 1, was this study approved by the IRB as a clinical study? If it is not a clinical study, please specify the study design.
Response: This is an interventional study. We added the word interventional for clarification in the abstract, the Institutional Review Board Statement and the Acknowledgement.
- Authors did not adress Comment 3. Providing brief information about the intervention is very important.
(Comment 3) I recommend authors to add historical, statistical (e.g. utilization) and institutional information (e.g. insurance) of Swedish massage
Response: We described the statistical information in response to your comment no 2 that the info has been given in the method section and that where was not additional insurance necessary.
In terms of historical information we added to the introduction a new chapter:
“The massage relates back to the Swedish physiologist and gymnastics instructor Pehr Henrik Ling (1776-1839), which subsequently has been promoted worldwide as Western or Swedish massage [3]. It has been used for the recovery of skeletal muscle from strenuous exercise [4,5], different chronic diseases [6], pain relief e.g. during labor [7], and balance performance [8].”
Further information is given in the method section:
“Swedish massage: Subjects participated in the morning between 8.00 and 11.00 in a single Swedish massage session lasting 45 minutes. The subjects carrying shorts laid supine on the massage table in a single booth in a warm room (22◦C) with background music. Typical treatments include effleurage (stroking), petrissage (kneading), tapotement (percussion), friction, and vibration. The massage was focused on the back, hamstrings, and calves to maximize muscle mass engagement and blood flow. Twenty-four massage therapy students (OSAO) performed the massage therapy under the close supervision of their teacher (registered under the National Supervisory Authority for Welfare and Health National Supervisory Authority for Welfare and Health operating under the Ministry of Social Affairs and Health), using consistent basic techniques and pressure. After 45 min, the subject stayed in the massage booth for an additional 60 min. No adverse events were recorded”.
Further information added to the ethics section: “Participants were insured by the University of Oulu.”
Reviewer 2 Report
Comments and Suggestions for Authors
Most of the remarks made have been adequately addressed in the revised manuscript, however on line 57 the sentence stll needs correcting.
Comments on the Quality of English LanguageQuality of English is okay.
Author Response
We thank the referee for time to review our manuscript. Please find our point to point response.
-Most of the remarks made have been adequately addressed in the revised manuscript, however on line 57 the sentence stll needs correcting.
Response: The changes were implemented accordingly.